

# The decay of the refocused Hahn echo in DEER experiments

Thorsten Bahrenberg[1#], Samuel M. Jahn[2#], Akiva Feintuch[1], Stefan Stoll[2], Daniella Goldfarb[1]

[1]Department of Chemical and Biological Physics, Weizmann Institute of Science, Rehovot, 7610001, Israel
[2]Department of Chemistry, University of Washington, Seattle, Washington 98195, USA

*Correspondence to*: Stefan Stoll (stst@uw.edu), Daniella Goldfarb (daniella.goldfarb@weizmann.ac.il)

\# - Equal contribution

**Abstract.** Double electron–electron resonance (DEER) is a pulse electron paramagnetic resonance (EPR) technique that measures distances between paramagnetic centres. It utilizes a four-pulse sequence based on the refocused Hahn spin echo. The echo decays with increasing pulse sequence length $2(\tau_1 + \tau_2)$, where $\tau_1$ and $\tau_2$ are the two time delays. In DEER, the
value of $\tau_2$ is determined by the longest inter-spin distance that needs to be resolved, and $\tau_1$ is adjusted to maximize the echo amplitude and thus sensitivity. We show experimentally that for typical spin centres (nitroxyl, trityl, Gd(III)) diluted in frozen protonated solvents, the largest refocused echo amplitude for a given $\tau_2$ is obtained neither at very short $\tau_1$ (which minimizes the pulse sequence length) nor at $\tau_1 = \tau_2$ (which maximizes dynamic decoupling for a given total sequence length), but rather at $\tau_1$ values smaller than $\tau_2$. Large-scale spin dynamics simulations including the electron spin and several hundred
neighbouring protons reproduce the experimentally observed behaviour almost quantitatively. They show that electron spin dephasing is driven by solvent protons via the flip-flop coupling among themselves and their hyperfine couplings to the electron spin.

## 1 Introduction

DEER (double electron–electron resonance) spectroscopy is a highly effective method for nanometer-scale distance
measurements in bio-macromolecules such as proteins, nucleic acids, and their complexes. (Jeschke, 2013)(Jeschke and Polyhach, 2007) This method measures the magnetic dipolar coupling between two spin labels attached at well defined, specific locations in the bio-macromolecules. As the dipolar coupling strength is inversely proportional to the cube of the inter-spin distance, the distance distribution between the two spin labels can be determined from the DEER signal.

The original implementation of the DEER experiment, as introduced by Milov et al. (Milov et al., 1984), is referred to as three-
pulse DEER (Fig. 1a). It employs the standard two-pulse (Hahn) echo at one frequency, $\nu_1$, and an additional pump pulse at another frequency, $\nu_2$, applied at time $t$ after the first pulse. The echo amplitude is measured as a function of the pump pulse position $t$. This sequence suffers from instrumental artefacts near $t = 0$ due to pulse overlap between the first pulse and the pump pulse. To eliminate these artefacts, the dead-time-free four-pulse DEER experiment was introduced by Spiess and co-workers (Fig. 1b).(Pannier et al., 2000) This method utilizes an additional refocusing $\pi$ pulse to generate the refocused echo





instead of the Hahn echo, allowing the undistorted measurement of the echo around $t = 0$. Again, the echo amplitude is

measured as a function of pump pulse position $t$. Currently, the four-pulse DEER sequence is the most widely used DEER

experiment.

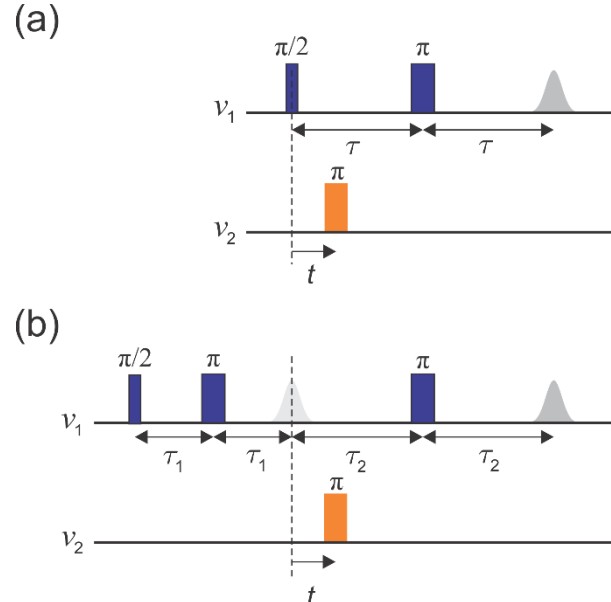

**Figure 1: Schematics of (a) the three-pulse DEER sequence and (b) the dead-time-free four-pulse DEER sequence. Observer pulses**
**are in blue, and pump pulses in orange. The dashed line indicates $t = 0$.**

The echo decays as the pulse sequence length increases, and the time scale of this decay is characterized by $T_M$, called the

phase memory time or decoherence time. Long $T_M$ values translate into high sensitivity. The longest possible pulse sequence

length that still gives a sufficient echo amplitude determines the longest distance that can be resolved. Therefore, a long $T_M$ is

required to access long distances.

The DEER signal-to-noise ratio (SNR) depends on several factors that for three-pulse DEER are approximately described by

(Zecevic et al., 1998)

$$SNR \propto V_0 \lambda \frac{e^{-(2\tau/T_M)^x}}{\sqrt{T_1}}, \tag{1}$$


where $V_0$ is the echo intensity at $\tau = 0$ (which is proportional to the number of spins in the sample); $\lambda$ is the modulation depth

and represents the fraction of spins inverted by the pump pulse; $x$ is a stretching exponent, and $T_1$ is the spin–lattice relaxation

time, which determines the rate of the data accumulation. Equation (1) is also applied in an approximate fashion for four-pulse

DEER by replacing $\tau$ with $\tau_1 + \tau_2$ and setting $x = 1$ (Jeschke and Polyhach, 2007).



According to Eq. (1), for a fixed evolution time the sensitivity decreases exponentially with decreasing $T_M$. Therefore, one strives to prolong $T_M$ as much as possible. It is possible to optimize the sample to suppress some of the mechanisms that contribute to dephasing. For example, the spin concentration can be lowered to minimize dephasing contributions due to electron–electron dipolar interactions, such as spectral and instantaneous diffusion(Eaton and Eaton, 2000, 2016; Raitsimring

et al., 1974). However, this concentration reduction leads to a loss in absolute signal intensity and may significantly prolong the experiment run time. Another mechanism that strongly contributes to dephasing is nuclear spin diffusion, which is driven by magnetic nuclei that are coupled to the electron spin and among themselves.(Brown, 1979; Canarie et al., 2020a; Huber et al., 2001; Lenz et al., 2017; Milov et al., 1972, 1973; Zecevic et al., 1998) The dephasing by this mechanism is enhanced in particular by nuclei with a large gyromagnetic ratio such as protons. Therefore, a reduction of the proton concentration leads

to a longer $T_M$. This can be partially achieved by using deuterated solvents, which is a common practice nowadays,(Jeschke and Polyhach, 2007) and—more completely but with more effort—by deuterating also the protein.(El Mkami et al., 2014; Schmidt et al., 2016) Additionally, the contribution of nuclear-spin-driven dephasing can be reduced by the application of dynamic-decoupling schemes such as the Carr–Purcell–Meiboom–Gill (CPMG) sequence(Carr and Purcell, 1954), thus prolonging $T_M$.(Harbridge et al., 2003) This concept was behind the design of the 5-(Borbat et al., 2013) and 7-pulse(Spindler

et al., 2015) DEER sequences, which have been shown to allow significantly longer evolution times and access to longer distances. These experiments, however, are not straightforward to run because of the contribution of unwanted transfer pathways that generate additional signal contributions to the DEER trace (Breitgoff et al., 2017). The effect of dynamic decoupling on the dephasing of paramagnetic centres in frozen solutions and in solids has been investigated in several detailed studies.(Kveder et al., 2019; Ma et al., 2014; Soetbeer et al., 2018)

A common practice for DEER is to record the two-pulse echo decay to estimate the maximum evolution time that can be applied for a particular sample. However, in the case of four-pulse DEER, a refocused two-pulse echo is observed and therefore it is the decay of this refocused echo which determines the SNR of the experiment and the distance accessibility. In the context of DEER, it is usually assumed that the refocused echo decays monotonically as a function of the overall pulse sequence length $2(\tau_1 + \tau_2)$, similar to the two-pulse echo. Based on this assumption, short $\tau_1$ values are chosen, to minimize phase relaxation

during this initial interval. In this work, we show that this approach is not generally optimal, particularly in protonated solvents. We explored the decay dependence of the refocused echo on $\tau_1$ and $\tau_2$, which is necessary for the optimization of the $\tau_1$ values for achieving the best four-pulse DEER SNR. For this, we used three types of spin probes commonly used in DEER experiments on proteins: a common nitroxide radical, a trityl radical, and a Gd(III) complex (see Fig. 2), all in dilute frozen aqueous solutions. We show that under conditions where the contribution of solvent nuclei to dephasing is significant (low

concentrations and low temperatures), the refocused echo sequence acts as a CPMG-like dynamic-decoupling sequence with two $\pi$ pulses.(Borbat et al., 2013) In this case, under the condition of a fixed and relatively long $\tau_2$, necessary for DEER measurements, we find the refocused-echo intensity reaches its maximum at relatively long $\tau_1$ values that are shorter than $\tau_2$. This is in contrast to the common use of a short $\tau_1$, chosen to minimize the overall pulse sequence length. We present numerical



spin dynamics simulations that reproduce the experimental results almost quantitatively and provide insight into the nature of

dynamic decoupling in the refocused Hahn echo.

![Figure 2 chemical structures]

**Figure 2. Chemical structures of the paramagnetic centres studied in this work: (a) 3-maleimido-proxyl, (b) trityl OXO63, and (c) Gd-C2.**

**2 Methods**

Sample preparation: 3-maleimido-proxyl and the trityl OXO63 were purchased from Sigma–Aldrich and from Oxford Instruments, respectively, and were used as provided. The powders were dissolved in either $H_2O$ or $D_2O$ to yield a 50 mM stock solution. They were further diluted using (a) a mixture of 80% water and 20% glycerol (v/v), (b) a mixture of 80% $D_2O$ and 20% glycerol-$d_8$, or (c) mixtures thereof to create 25%, 50%, or 75% deuterated samples. The final radical concentration

was 100 µM in all cases.

GdCl$_3$ was purchased from Sigma–Aldrich and used at a final concentration of 100 µM. The protein MdfA labelled with Gd-C2 at positions 44 and 307 and solubilized in detergent was prepared earlier according to a published protocol(Yardeni et al., 2019) and used without further modification. The final concentration of MdfA was about 25 µM.

Spectroscopic measurements: All measurements were carried out on a W-band (94.9 GHz) home-built spectrometer.(Goldfarb

et al., 2008) (Mentink-Vigier et al., 2013) Two-pulse echo decays were recorded using the sequence shown in Fig. 1a without the pump pulse utilizing a two-step phase cycle. Two-dimensional refocused echo experiments were recorded using the sequence given in Fig. 1b without the pump pulse, and the echo intensity was measured as a function of both $\tau_1$ and $\tau_2$. An 8-step phase cycle (x)(x)(x) was used (+/−x on all three pulses). All experimental parameters are listed in Table 1. In each case, the magnetic field was set to a value where the maximum of the EPR spectrum was resonant with the microwave frequency.

To produce the final signal, the echo was integrated over its full width at half maximum.

**Table 1. Experimental parameters used in this work. Identical parameters were used for the two-pulse and refocused echo measurements.**





| Sample | $\pi/2$ pulse (ns) | $\pi$ pulse (ns) | Repetition time (ms) | Temperature (K) |
|---|---|---|---|---|
| 3-maleimido-proxyl | 25 or 20 | 50 or 40 | 20 | 25 |
| trityl OXO63 | 25 or 20 | 50 or 40 | 100 | 25 |
| Gd(III)[a] | 15 | 30 | 0.3 | 10 |

[a] For both $GdCl_3$ and Gd-C2 labelled MdfA.


Simulations: To simulate the refocused echo decay, we follow our previously published approach.(Canarie et al., 2020a) A 3-maleimido-proxyl radical was geometry-optimized using density functional theory (DFT, B3LYP, def2-SVP) and then solvated in a periodic box containing $H_2O$/glyercol (3038 waters, 188 glycerols) using molecular dynamics. The spin system used in the spin dynamics simulation includes the unpaired electron on the radical, all protons on the radical, as well as all protons from $H_2O$ and glycerol molecules within 12 Å of the electron spin (512 protons total). The spin Hamiltonian includes full nucleus–nucleus coupling tensors as well as the secular and pseudosecular parts of all hyperfine coupling tensors, calculated from the electron and nucleus positions. The echo decay was simulated by explicit fully coherent time evolution of the spin system state using density matrix propagation in Hilbert space, without any explicit relaxation terms. The calculations were performed using a truncated ensemble cluster correlation expansion (CCE)(Yang and Liu, 2008, 2009), which is a refinement of the earlier cluster expansion. (Witzel and Das Sarma, 2006) Echo signals from all possible subsystems involving the electron spin and a small cluster of nuclei are calculated separately, and the resulting signals are combined to give the total signal. To limit the number of clusters, we used a maximum nuclear-cluster size of 2, 3, or 4 (2-CCE, 3-CCE, 4-CCE), and neglected all clusters with intra-cluster nucleus–nucleus couplings smaller than 1.58 kHz. This gave a converged signal and included 4365 two-proton clusters and 52,937 three-proton clusters. Orientational averaging was performed over a Lebedev grid with 14 points, taking the orientation-dependent excitation efficiency of the pulses into account. The simulations used an applied magnetic field of 3.38 T.

**3. Results and Discussion**

To investigate the refocused echo decay, we acquired the refocused echo amplitude as a function of both $\tau_1$ and $\tau_2$ for frozen solutions of a 3-maleimido-proxyl radical, the trityl OXO063, and Gd(III) (see Fig. 2). We measured all samples in both $H_2O$/glycerol (80:20, v/v) and $D_2O$/glycerol-$d_8$ (80:20, v/v) to examine the effect of nuclear spin diffusion. This is known to be a dominant mechanism of dephasing for organic radicals in dilute frozen solutions at cryogenic temperatures(Canarie et al., 2020b; Eaton and Eaton, 2000; Zecevic et al., 1998)and has been shown to be partially suppressed using dynamic decoupling(Harbridge et al., 2003).





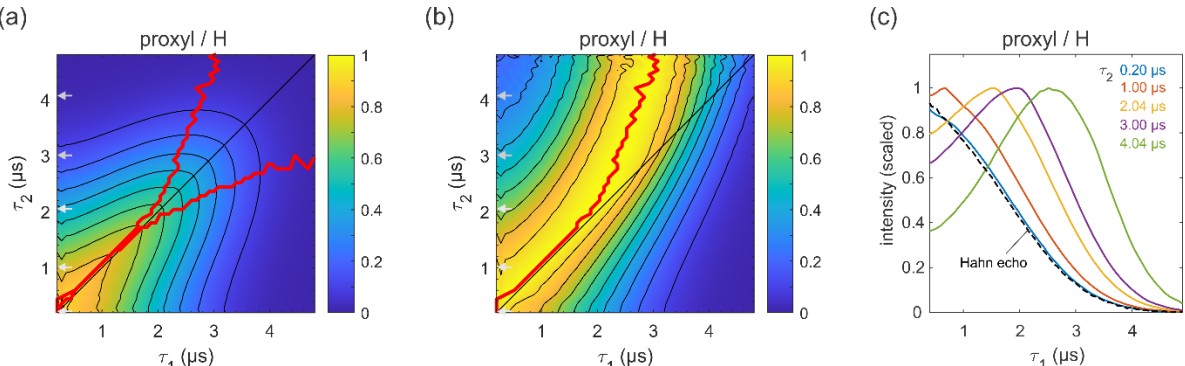


**Figure 3: Refocused-echo decay for 100 μM 3-maleimido-proxyl in H$_2$O/glycerol (80:20 v/v) at 25 K. Panel (a) shows the echo amplitude as a function of $\tau_1$ and $\tau_2$, and (b) shows the same data after normalization of each slice along $\tau_1$. The red lines in (a) and (b) indicate the location of the maxima along $\tau_1$ for fixed $\tau_2$ (upper line), and vice versa (lower line, only in (a)). Panel (c) shows slices along $\tau_1$ for several $\tau_2$ values (indicated by grey arrows in (a) and (b)), together with a Hahn echo decay.**


### 3.1 Organic radicals

Fig. 3a presents the measured refocused echo decay data of 3-maleimido-proxyl in H$_2$O/glycerol. It shows an overall monotonic decay as the values of $\tau_1$ and $\tau_2$ increase and decay is symmetric with respect to exchange of $\tau_1$ and $\tau_2$. The data show that the echo decay is not a function of only the total pulse length, $2(\tau_1 + \tau_2)$, in contrast to the two-pulse echo decay.

The decay is fastest along $\tau_1 \approx 0$ and $\tau_2 \approx 0$, and slowest along the diagonal $\tau_1 = \tau_2$. The latter is a manifestation of CPMG dynamic decoupling. While these general features of the decay are at least qualitatively as expected, the decay shape has a more subtle feature that has important practical implications for DEER measurements. In DEER, the choice of $\tau_2$ is determined by the desired distance range, and optimal choice of $\tau_1$ maximizes the echo and thereby SNR. Optimizing $\tau_1$ for a fixed $\tau_2$ corresponds to finding the maximum along a particular horizontal slice of the data in Fig. 3a. This is more obvious in Fig. 3b,

which shows the data from Fig. 3a after individual normalization of each horizontal $\tau_1$-dependent trace with fixed $\tau_2$. The superimposed red curve represent the loci of the maxima along these slices, corresponding to optimal $\tau_1$ values for DEER. A similar plot and curve can be generated for vertical slices (fixed $\tau_1$, variable $\tau_2$), due to the symmetry of the decay across the $\tau_1 = \tau_2$ diagonal. The curves are also superimposed in Fig. 3a. The crucial feature of theses curves is their deviation from the diagonal ($\tau_1 = \tau_2$) for increasing values of $\tau_1$ and $\tau_2$.

Fig. 3c shows individual slices at several $\tau_2$ values (indicated in Fig. 3a and 3b by grey arrows) together with a two-pulse echo decay (dashed line) as reference. This shows that in protonated solvents the refocused echo decay along $\tau_1$ (and, by symmetry, along $\tau_2$) cannot generally be described by a stretched exponential decay. While there is little difference between the refocused echo decay and the two-pulse echo decay for small $\tau_2$ values, for larger $\tau_2$ values the refocused echo first grows significantly from its short-$\tau_1$ amplitude, forms a broad maximum, and only then starts decaying. As a general trend, the longer $\tau_2$, the more

pronounced is the effect. This is relevant for DEER experiments, which are usually run with long $\tau_2$ values to provide long



dipolar evolution times. Interestingly, the maximum intensity does not appear at $\tau_1 = \tau_2$, where it would match the CPMG condition of maximal dynamic decoupling. Nor does it occur at $\tau_1 \approx 0$, corresponding to the minimal total pulse sequence length. Rather, it is slightly 'detuned' from the CPMG diagonal and appears at $\tau_1 < \tau_2$. This deviation from the diagonal increases with increasing $\tau_2$ values. In practice, for a DEER experiment in protonated solvents, this shows that setting $\tau_1$ to a

short value (to minimize total pulse sequence length), or equal to $\tau_2$ (to maximize dynamic decoupling), leads to a significant and unnecessary loss of sensitivity.

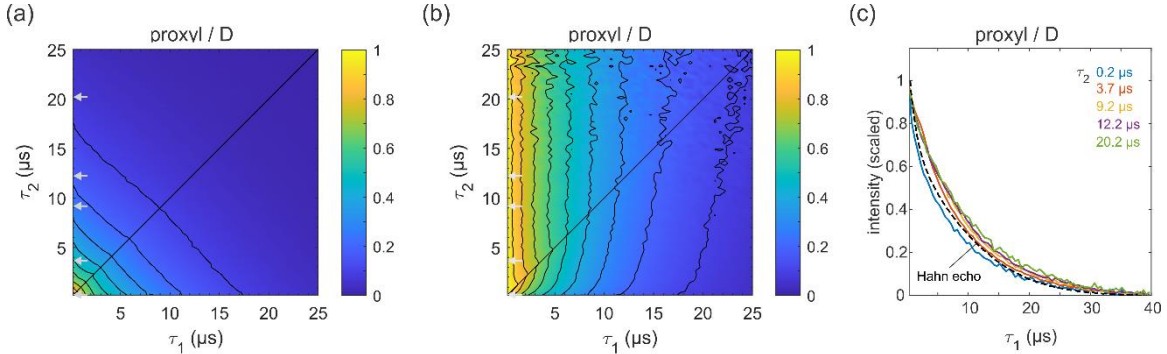

**Figure 4: Refocused-echo decay for 100 μM 3-maleimido-proxyl in D₂O/glycerol-d₈ (80:20 v/v) at 25 K. Panel (a) shows the echo**
**amplitude as a function of $\tau_1$ and $\tau_2$, and (b) shows the same data after normalization of each slice along $\tau_1$. Panel (c) shows slices along $\tau_1$ for several $\tau_2$ values (indicated by grey arrows in (a) and (b)), together with a Hahn echo decay.**

Figure 4a shows the refocused-echo decay for 3-maleimido-proxyl in the corresponding deuterated solvent. Here, both the Hahn echo and the refocused-echo decays are significantly extended as compared to the protonated solvent, as expected, due

to the absence of protons in the sample. The effect observed in protonated solvent is entirely absent here. The echo amplitude depends only on the total pulse sequence length, $2(\tau_1 + \tau_2)$. The decays along $\tau_1$ for fixed $\tau_2$ are essentially independent of $\tau_2$ and resemble very much the two-pulse echo decay (see Fig. 4b,c). It is apparent that nuclear spin diffusion is suppressed here and dynamic decoupling is ineffective. The decay is dominated by other dephasing mechanisms such as instantaneous diffusion and therefore, a short $\tau_1$ gives optimal sensitivity.






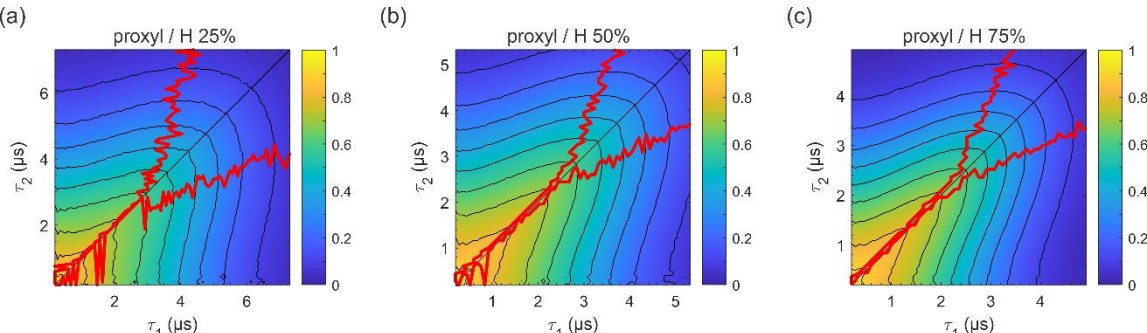

**Figure 5: Refocused-echo decay for 100 µM 3-maleimido-proxyl in H₂O/glycerol solvents (80:20 v/v) with varying degrees of solvent protonation at 25 K, (a) 25%, (b) 50%, and (c) 75%. The red lines indicate the location of the maxima along $\tau_1$ for fixed $\tau_2$ (upper line), and vice versa (lower line). Plots for 100% and 0% solvent protonation are shown in Figures 3a and 4a, respectively.**

In order to evaluate how the degree of protonation affects the refocused-echo decay shape, we measured samples in partially deuterated solvent with 25%, 50%, and 75% protonation. The results are shown in Fig. 5. These measurements reveal that already at 25% protonation there is a strong impact on the decay time scale and shape, as can be seen by comparing Fig. 5a to Fig. 4a. At 50% protonation, the effect is close to complete (Fig. 5b and Fig. 5c compared to Fig. 3a). The $T_M$ values determined from two-pulse echo decays, given in Table 2, show a similar trend. They decrease with increasing proton concentration. The impact is strongest when going from 0% protonation to 25% and 50% protonation. The differences between 50%, 75% and 100% protonation are smaller.

**Table 2: The phase memory time, $T_M$, and the stretching exponent, $x$, obtained from fitting stretched exponential decays to the two-pulse echo decays (25 K) of the 3-maleimido-proxyl samples with different degrees of solvent protonation.**

| % Protonation | $T_M$, µs | $x$ |
|---|---|---|
| 100 | 2.14(1) | 2.01(2) |
| 75 | 2.72(1) | 2.11(2) |
| 50 | 3.31(2) | 1.99(2) |
| 25 | 4.29(2) | 1.91(3) |
| 0 | 6.4(1) | 0.65(1) |



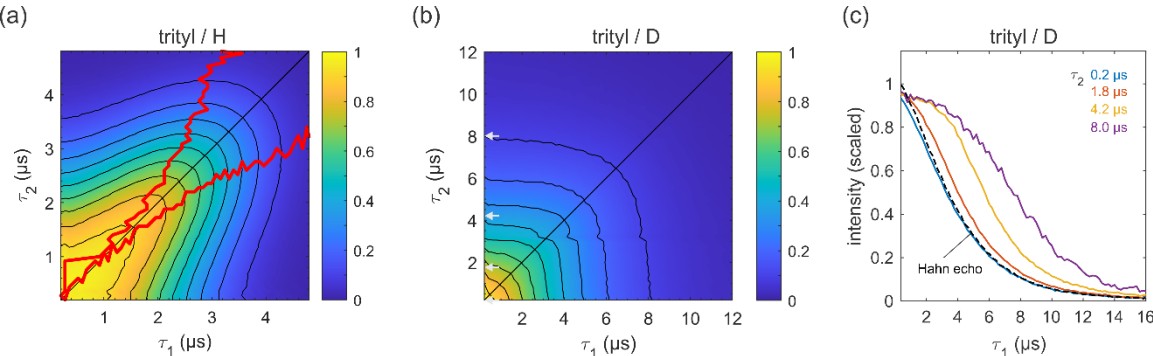

**Figure 6: Refocused-echo decays for 100 μM trityl OXO63 in (a) H₂O/glycerol (80:20 v/v) and (b) D₂O/glycerol-d₈ at 25 K; in panel (a), the red lines show the position of the echo maximum along $\tau_1$ for each constant $\tau_2$ (upper line), and vice versa (lower line). Panel (c) shows slices of the data in panel (b) along $\tau_1$ for several $\tau_2$ values (indicated by grey arrows in (b)), together with a Hahn echo decay.**

Figure 6 shows refocused-echo decay data of the trityl OXO63, which is also used as a spin label for DEER on proteins and nucleic acids(Yang et al., 2012)(Reginsson et al., 2012)(Giannoulis et al., 2019). The data reveal the same behaviour as observed for 3-maleimido-proxyl. Again, in fully protonated solvent (Fig. 6a), the decay is fastest along $\tau_2 \approx 0$ and $\tau_1 \approx 0$, and it is slowest along the diagonal $\tau_1 = \tau_2$. The time scale is very similar to that of the 3-maleimido-proxyl decay. As for 3-maleimido-proxyl, the location of the slice-wise maximum echo intensities (indicated by red lines) again deviates from the CPMG diagonal as $\tau_2$ (or $\tau_1$) increases. In contrast, in fully deuterated solvent (Fig. 6b), no maxima in the echo intensities are observed, although there is a clear deviation from a pure $(\tau_1 + \tau_2)$-dependent decay as compared to 3-maleimido-proxyl in deuterated solvent (Fig. 4). Fig. 6c clearly shows that as $\tau_2$ increases, the echo intensity as a function of $\tau_1$ persists longer. This indicates that nuclear spin diffusion, induced by the trityl OXO63 protons themselves, is still a contributing mechanism, though not dominant enough to make CPMG dynamic decoupling effective along $\tau_1$.

## 3.2 Gd(III)

Next, we carried out similar measurements on high-spin Gd(III) ($S$=7/2), as Gd(III)–Gd(III) DEER on proteins is becoming more common.(Feintuch et al., 2015) The results for GdCl₃ in protonated and deuterated solvents are shown in Fig. 7. In both solvents, the decay is essentially symmetric with respect to $\tau_1$ and $\tau_2$. In protonated solvent (Fig. 7a), the shape of the 2D decay is similar to the ones observed for 3-maleimido-proxyl and trityl OXO63, with slice-wise echo maxima detuned from the CPMG condition (red lines). For small $\tau_2$ values below about 1.5 μs, the optimal $\tau_1$ is as short as possible, indicating that a second dephasing mechanism in addition to nuclear spin diffusion is contributing, such as the transient zero field splitting mechanism.(Raitsimring et al., 2014) This is quite different from the organic radicals. In deuterated solvent (Fig. 7b), the decay





more closely resembles those observed for the organic radicals (Figs. 4a and 6b). As evident from Fig. 7c, nuclear spin diffusion plays a role in dephasing that is lower than in trityl OXO63 (Fig.6c) but higher than in 3-maleimido-proxyl (Fig. 4c).

In all cases studied a particularly interesting observation is the deviation of the maximum echo from the diagonal (the red lines in Figs. 3a,5a, 6a and 7a). For 3-maleimido-proxyl, the point at which the maxima starts to deviate from the diagonal starts around $\tau_1 = 1.5$ μs for 100% protonation and increases to about 2.7 μs for 25% protonation. The deviation point for trityl OXO63is similar, but for Gd(III) the behavior is different, the maximum echo is never observed along the diagonal, even for short $\tau_1, \tau_2$. This general non-monotonic behaviour contrasts with the usual assumption of a monotonic decay and is 230 rationalized as a balance between two competing effects that occur as $\tau_1$ is increased from 0 to $\tau_2$: increased dephasing due to extending the pulse sequence length on one hand, and the gain in amplitude due to increasing degree of dynamic decoupling as $\tau_1$ approaches $\tau_2$ on the other hand.

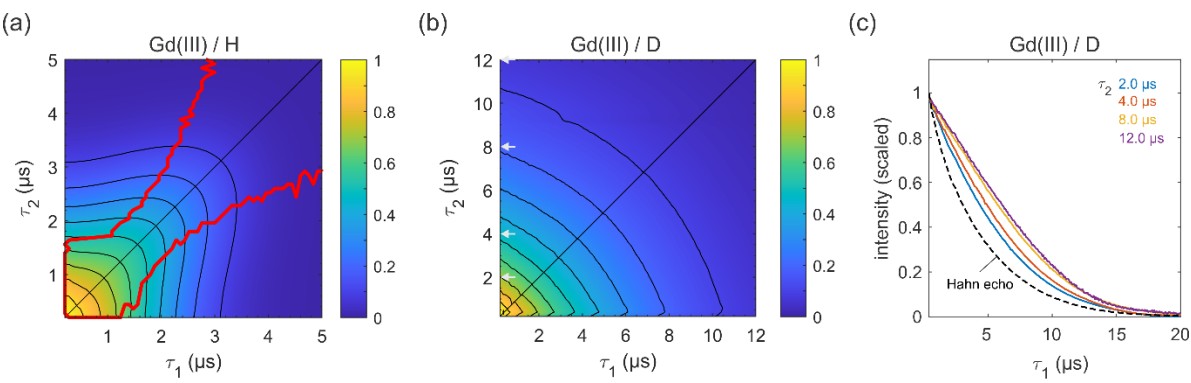

**Figure 7: Refocused-echo decays for 100 μM GdCl₃ in (a) H₂O/glycerol (80:20 v/v) and (b) D₂O/glycerol-d₈ at 10 K; in panel (a), the red lines show the position of the echo maximum along $\tau_1$ for each constant $\tau_2$ (upper line), and vice versa (lower line). Grey arrows in (b) indicate the $\tau_2$ values of slices shown in (c) together with a Hahn echo decay.**

Most applications of DEER are carried out on samples with deuterated solvent, typically a mixture of D₂O/buffer and glycerol-240 d₈. However, complete deuteration is rarely achieved, as the buffer, substrates, detergent, lipid membrane, and the protein or nucleic acid of interest contain non-exchangeable protons that can be in close proximity to the spin label. There are also cases where proteins precipitate in D₂O. (Verheul et al., 1998)(Reslan and Kayser, 2018) As it is now clear that the echo decay is strongly influenced by the presence of even small amounts of protons in the sample, it is still advisable to optimize the value of $\tau_1$ even if the sample is only partially protonated. An example for a sample with incomplete deuteration is the protein MdfA 245 V44C/V307C doubly labelled with the Gd-C2 tag. MdfA is protonated and, being a membrane protein, is solubilized in detergent (n-dodecyl-β-D-maltopyranoside, DDM) micelles, which have a long alkyl chain with non-exchangeable protons. Consequently, even at 100% solvent deuteration, a significant fraction of protons is present in the sample. Fig. 8 shows the measured refocused echo decay as a function of $\tau_1$ for selected fixed $\tau_2$ times. The non-monotonic behavior at the largest $\tau_2$ is qualitatively similar to that observed for the organic radicals and GdCl₃. While the maximum is not as pronounced, this





behaviour still has practical implications: The dipolar evolution time of DEER is typically in the range of 3–4 µs (blue trace), so $\tau_1$ can safely be extended to almost 3 µs without loss of sensitivity. This was exploited, for example, to remove instrumental artefacts from the DEER trace when shaped pulses are used. (Bahrenberg et al., 2019)

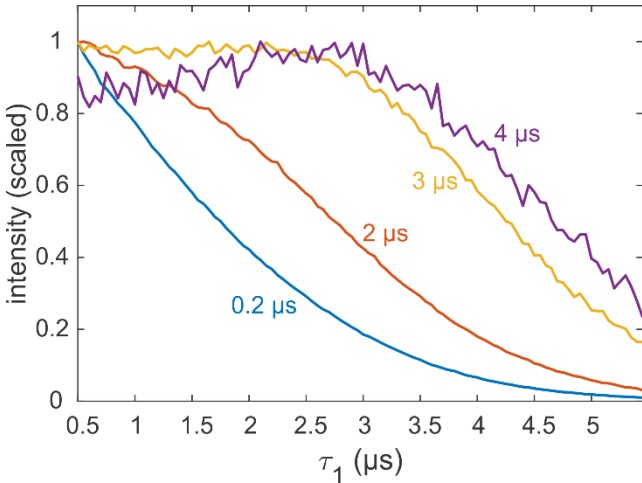

**Figure 8: Refocused-echo decay experiments (constant $\tau_2$, variable $\tau_1$) on 25 µM MdfA double mutant V44C/V307C doubly labelled with Gd-C2 at 10 K. Selected values for $\tau_2$ in ns are color-coded. Data taken from the SI of ref** (Yardeni et al., 2019)**.**

### 3.3 Simulations

To better understand the physical origin of the observed refocused-echo decay, we performed a numerical quantum spin
dynamics simulation for a 3-maleimido-proxyl radical solvated in $H_2O$/glycerol as a representative of the behaviour observed experimentally. The molecular and solvation geometries were determined by DFT and molecular dynamics, respectively. The fully coherent spin dynamics simulation included the unpaired electron on the radical and all 512 protons within 12 Å of the unpaired electron. All hyperfine and nucleus–nucleus coupling terms were included in the spin Hamiltonian. To handle the large Hilbert space with $2^{513}$ spin states, a truncated ensemble correlated cluster expansion (CCE) approach was utilized (see
Methods section). This methodology has recently been shown to accurately reproduce the two-pulse echo decays of organic radicals in frozen, dilute protonated solutions (Canarie et al., 2020a).





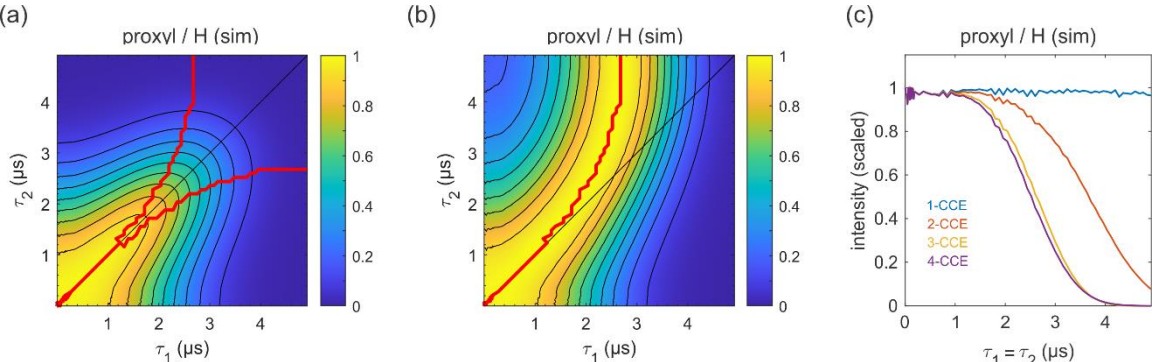

**Figure 9: Simulation of the refocused-echo decay for 3-maleimido-proxyl in $H_2O$/glycerol (80:20 v/v). Panel (a) shows the simulated**
**refocused-echo amplitude as a function of $\tau_1$ and $\tau_2$, using 2-CCE. The $\tau_1 = \tau_2$ line is shown in black. The upper red curve in panel**
**(a) indicates the ridge of panel (b), which normalizes each slice along $\tau_1$ (with constant $\tau_2$) to unit maximal amplitude. The lower**
**red curve in panel (a) is the analogous ridge for normalization along $\tau_2$. Panel (c) shows the simulated refocused-echo decay for $\tau_1 =$**
**$\tau_2$ as a function of cluster truncation level (1-CCE through 4-CCE).**

The simulated refocused-echo decay is shown in Fig. 9a and 9b. Remarkably, it almost quantitatively matches the experimental result (Fig. 3a and 3b) both in shape and time scale, in particular the deviation of the maxima from the diagonal for long $\tau_1$ and $\tau_2$. The simulations show that the only terms in the spin Hamiltonian affecting the decay are the secular parts of the hyperfine couplings and the flip–flop terms of the nucleus–nucleus coupling (data not shown). In contrast, if the nucleus–nucleus couplings is neglected, the effect disappears. This is shown in Fig. 9c: If only one-nucleus clusters are included, namely, all nucleus–nucleus couplings are neglected, no decay is seen. The shallow modulations are due to pseudosecular components of the hyperfine couplings. Including two-nucleus clusters in the simulation yields an echo decay that has the correct shape and an almost correct time scale. Adding three-nucleus clusters improves the time scale slightly, and including larger clusters does not lead to further improvements.

The conceptual essence of the mechanism can be pictured with one electron and a pair of nuclei, all coupled among each other. The associated three-spin system has eight eigenstates, with four nuclear eigenstates in each of the two electron spin manifolds ($\alpha$ and $\beta$). Due to presence of the flip–flop term of the nucleus–nucleus coupling, the nuclear eigenstates in the $\alpha$ manifold are different from those in the $\beta$ manifold (assuming the two hyperfine couplings are not identical), and formally forbidden EPR transitions with $\Delta m_I \neq 0$ have non-zero transition amplitudes. Excitation of the system from one of its eigenstates in one manifold into the other manifold therefore generates nuclear coherence. This results in a periodic modulation of the electron spin echo amplitude as a function of inter-pulse delays, in a fashion analogous to electron spin echo envelope modulation (ESEEM). (Schweiger and Jeschke, 2001)(Van Doorslaer, 2017) Every cluster of nuclei contributes such a periodic modulation to the overall echo, with varying amplitudes and frequencies depending on the structure of the cluster and its location relative to the electron spin. The solvent environment of the electron spin on the radical contains many nuclear clusters, and the echo modulations from all clusters combine to give an overall echo decay. Although the echo decays, the coherence lives on until it is destroyed by electron and nuclear $T_1$ relaxation processes.



Traditionally, this dephasing mechanism has been explained in terms of a stochastic nuclear spin diffusion model that involves flip-flop events between pairs of nuclei with a phenomenological flip–flop rate constant. (Milov et al., 1973; Zecevic et al., 1998) However, the first-principles simulation shown earlier (Canarie et al., 2020a) and here reveals that the term "diffusion" might not be entirely appropriate, as the quantum model that reproduces the echo decays is fully coherent and does not contain any relaxation terms or other stochastic elements. It might therefore be conceptually more accurate to refer to this dephasing mechanism as "nuclear-spin-cluster-driven electron spin decoherence", although this is clearly more tedious.

The dynamic decoupling effect in the refocused-echo decay, i.e. the decoherence suppression along $\tau_1 = \tau_2$, can be understood a little better with the simple model Hamiltonian:

$$\hat{H} = \mu_B g_e B_0 \hat{S}_z + \hbar \sum_n \left( -\mu_N g_n B_0 \hat{I}_{zn} + A_n \hat{S}_z \hat{I}_{zn} \right) + \hbar \sum_{m<n} b_{mn} \left( \hat{I}_{+m} \hat{I}_{-n} + \hat{I}_{-m} \hat{I}_{+n} - 4 \hat{I}_{zm} \hat{I}_{zn} \right), \tag{2}$$

where all symbols have their usual meaning. In particular, $A_n$ represents the secular hyperfine coupling of nucleus $n$. The last sum contains the secular and flip–flop components of the couplings between nuclei $m$ and $n$, with coupling parameter $b_{nm}$. Calculating the refocused echo amplitude $V(\tau_1, \tau_2)$ using density matrix propagation with ideal pulses, and Taylor expanding the resulting expression in terms of $b_{mn}$ gives a power series where the lowest-order non-vanishing term is second order in $b_{mn}$:

$$\langle V_2 \rangle = -\sum_{m,n} \frac{2 b_{mn}^2}{\omega_{mn}^2} \left[ \cos(\omega_{mn} \tau_1) - \cos(\omega_{mn} \tau_2) \right]^2, \tag{3}$$

where $\omega_{mn} = (A_m - A_n)/2$. This term is negative for $\tau_1 \neq \tau_2$, and zero for $\tau_1 = \tau_2$. $\langle V_2 \rangle$ also occurs as a factor to all higher-order terms in the power series. Factoring these out gives the power series for the exponential function, so that $\langle V_2 \rangle$ contributes a factor of $e^{\langle V_2 \rangle}$ to the overall echo amplitude. In general, each term in the Taylor series can be factored into terms that can be collected into exponentials, yielding

$$V(\tau_1, \tau_2) = e^{\langle V_2 \rangle + \langle V_3 \rangle + \cdots} = e^{\langle V_2 \rangle} \cdot f(\tau_1, \tau_2), \tag{4}$$

where $\langle V_3 \rangle$ is the term cubic in $b_{mn}$ and we have collected all terms of higher order in $b_{nm}$ than 2 into a single term $f$. This factorization is referred to as the linked-cluster expansion.(Saikin et al., 2007)

Under the CPMG condition of $\tau_1 = \tau_2$, $\langle V_2 \rangle$ vanishes, and therefore $e^{\langle V_2 \rangle} = 1$. The third-order term $e^{\langle V_3 \rangle}$ is negligible, leaving the fourth-order term as the lowest non-trivial term. (Ma et al., 2014) Deviating from the $\tau_1 = \tau_2$ line introduces a non-zero negative second-order term, and therefore $e^{\langle V_2 \rangle} < 1$. The significant three-spin terms are at least fourth-order in $b_{nm}$, and so are less significant when the second-order term contributes. This partially explains why 3-clusters in the simulations compress




$V(\tau_1, \tau_2)$ along the $\tau_1 = \tau_2$ line: the CPMG effect breaks down at the same order that the 3-clusters start to contribute. (Ma et al., 2014)

Maximizing echo intensity at a given $\tau_2$ requires determining a $\tau_1$ value that balances the loss from deviating from the CPMG condition (quadratic in $b_{nm}$) with the loss from a long total experiment (quartic in $b_{nm}$).

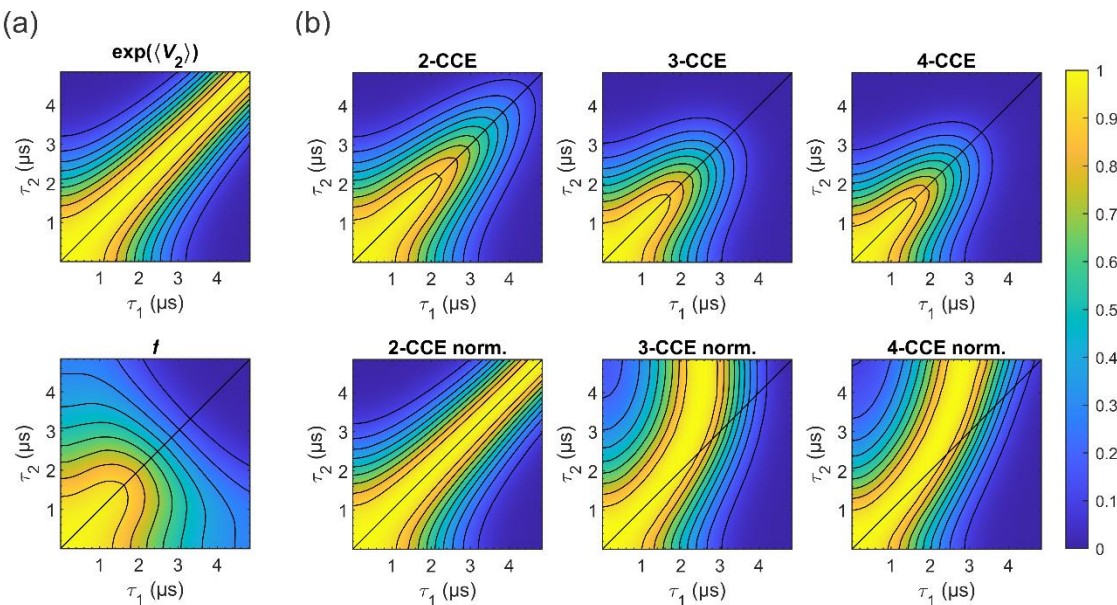

**Figure 10: Simulated refocused-echo decay of 3-maleimido-proxyl. (a) Factorization of the four-cluster decay into a term second**
**order in $b_{nm}$ (top) and all higher-order terms (bottom). (b) Simulated refocused-echo decays at two-, three-, and four-cluster level (2-CCE, 3-CCE, 4-CCE) (top), with the corresponding slice-wise normalized decays (bottom). For this simulation, a single orientation of the radical was used.**

Figure 10 illustrates the two factors $e^{\langle V_2 \rangle}$ and $f$ for the 3-maleimido-proxyl case, as well as CCE simulations up to 4-CCE in
both standard and slice-wise normalized forms. It can be seen that $e^{\langle V_2 \rangle}$ accounts for most of the CPMG effect (Fig.10a top), that $f$ represents predominantly the decay along $\tau_1 + \tau_2$ (Fig. 10a bottom), and that 3-nucleus clusters contribute most to the part of $f$ that drive the slice-wise maximum away from $\tau_1 = \tau_2$ (slice-wise diagrams, Fig. 10b bottom).

Using the calculated shapes of $e^{\langle V_2 \rangle}$ and $f$ in Fig. 10, we can rationalize the existence of a maximum of $V(\tau_1, \tau_2)$ along $\tau_1$ for fixed $\tau_2$, i.e. why we observe $\partial V / \partial \tau_1 = 0$ for some $0 < \tau_1 < \tau_2$. The derivative is

$$\frac{\partial V(\tau_1, \tau_2)}{\partial \tau_1} = e^{\langle V_2 \rangle} \frac{\partial f}{\partial \tau_1} + V(\tau_1, \tau_2) \frac{\partial \langle V_2 \rangle}{\partial \tau_1}. \tag{5}$$


At $\tau_1 = \tau_2$, both derivatives on the right-hand side are negative (as seen in Fig. 10), rendering $\partial V / \partial \tau_1$ negative – increasing $\tau_1$ beyond $\tau_2$ always decreases the echo, since the CMPG suppression is lost *and* the total evolution time gets longer. In



addition, for a given $\tau_2$ and for some region $\tau_1 < \tau_2$, $e^{\langle V_2 \rangle}$ grows more rapidly with $\tau_1$ than $f$ decays with $\tau_1$, as seen in Fig. 10. Therefore, $\partial V / \partial \tau_1 > 0$, which indicates that there is at least one $\tau_1$ for which an increase in $\tau_1$ increases $V$. This is related to the Taylor series converging fast enough. An increase of $V$ with $\tau_1$ indicates the gain in echo amplitude from approaching the CPMG condition is larger than the loss from prolonging the total evolution time. Taken together, this means that $\partial V / \partial \tau_1$ must be zero for some $\tau_1 < \tau_2$, and therefore $V$ maximal. At that point, the effects of increasing $e^{\langle V_2 \rangle}$ (signal gain due to better dynamic decoupling) and decreasing $f$ (signal loss due to longer evolution time) are balanced.

## 4. Conclusions

We observe that for low-concentration nitroxide, trityl and Gd(III) paramagnetic centres in protonated solvents, where the socalled nuclear spin diffusion decoherence mechanism dominates, the refocused two-pulse echo amplitude as a function of $\tau_1$ for a fixed $\tau_2$ was maximal not for $\tau_1 = 0$ (which minimizes total pulse sequence duration) nor $\tau_1 = \tau_2$ (which maximizes dynamic decoupling given a fixed total pulse sequence duration), but rather for a $\tau_1$ value between 0 and $\tau_2$. We observed this effect in samples with 25–100% solvent protonation. In fully deuterated solvents, the effect was not observed owing to fact that other dephasing mechanisms (such as instantaneous diffusion) become significant or dominant, at least at the concentrations employed in this study.

First-principles spin dynamics simulations using a solvated nitroxide radical structure accurately reproduced both the time scale and the shape of the observed refocused-echo decay, indicating that it is due to the large number of protons proximal to the spin label. This confirms that nuclear-spin-driven decoherence is the main mechanism of echo decay in the protonated samples under the conditions investigated (low concentration, low temperature).

These findings have practical implications for DEER experiments. There, $\tau_2$ is typically determined by structural considerations such as the longest distances that needs to be resolved. The choice of $\tau_1$ that maximizes SNR for a given $\tau_2$ is therefore important. Our results indicate that it is important to explore the entire range of possible $\tau_1$ values in order to find the maximum SNR for the DEER measurement, in particular for samples that cannot be produced with 100% deuteration of all components (solvent, protein, detergent, etc.).

## Code and data availability

All measured data are available at doi:10.5281/zenodo.4449018



**Author contributions**

TB, AF, and DG initiated the project; TB, SMJ, AF, SS, and DG designed research; TB performed all experiments and analysed the data; SMJ implemented the code and performed all simulations; all authors wrote the manuscript.

**Competing interests**

The authors declare that they have no competing interests.

**Acknowledgements**

This research was in part supported by the National Institutes of Health (R01 GM125753; S.S. and S.M.J.) and by the Minerva Foundation (D.G.). This research was made possible in part by the historic generosity of the Harold Perlman Family (D. G.). D. G. holds the Erich Klieger Professorial Chair in Chemical Physics.

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
