# Peer review of "The decay of the refocused Hahn echo in DEER experiments"

_Magnetic Resonance, 2021_

## Author Comment (AC1)

**Comment 1**

Below we copied the comments in black and our response is in red.

The authors of the manuscript "The decay of the refocused Hahn echo in DEER experiments" under discussion at Magnetic Resonance experimentally test optimal DEER observer sequence settings for two refocusing pulses (*i.e.* optimizing the 4- or 5-pulse DEER scheme) for a nitroxide, a trityl radical and a gadolinium(III) ion in frozen protonated and deuterated water-glycerol glass. CCE simulations for the nitroxide in a water-glycerol mixture rationalize the observed decoherence behavior for cases where proton-driven nuclear spin diffusion induces electron spin decoherence. Overall, the Results and Discussion section on the experimental data would benefit from a more careful and consistent discussion on contributing dephasing mechanisms. In this context, I would like to draw the authors' attention to our own article called "*Dynamical decoupling in water-glycerol glasses: a comparison of nitroxides, trityl radicals and gadolinium complexes*" under review at another journal since January 6$^{th}$ 2021. We made this manuscript available as a preprint under https://doi.org/10.26434/chemrxiv.13678447.v1 (hereinafter called DD-watergly2021) to facilitate the discussion here.

We were not aware of the results in this preprint (abbreviated henceforth as JS21), as it appeared on Chemrxiv after we submitted our manuscript to Magn.Reson., so unfortunately we could not refer to its findings in our initial submission. Indeed it is relevant to our manuscript, and we now cite it in the appropriate places.

We would like to emphasize that our work has a different scope than JS21. Our work does not aim at analyzing all dephasing mechanisms of paramagnetic centers in deuterated and protonated solvents, but rather focuses on the refocused echo decay in the context of DEER, and on how to optimize its parameters for the practically relevant cases where protons have a major contribution to the decoherence, due to their presence on proteins or on detergent molecules, which are practically expensive to deuterate.

For this, we present a theoretical analysis of the nuclear-spin-bath driven decoherence mechanism that is fully based on first principles, which is praised by the three reviewers. This is different from the approach used in JS21, which considers more mechanisms, but on a purely phenomenological basis.

**Comments regarding the discussion on dephasing mechanisms:**

- **Fig. 3/157-159:** "While there is little difference between the refocused echo decay and the two-pulse echo decay for small $\tau_2$ values." I agree with this observation, though the authors could strengthen the interpretation of their results by also noting the progressive change of the maxima along $\tau_1$ for increasing $\tau_2$ in Fig. 3c. The

maximum at $\tau_2$ = 1 μs is particularly sharp and broadens for $\tau_2$ > 1 μs. This effect and the described deviation originate from the "fast" decoherence process driven by nitroxide methyl nuclei at low temperatures first demonstrated in glassy *o*-terphenyl (Soetbeer et al., 2018) which also contributes in water-glycerol glass (see DD-watergly2021: Fig. 4b) at the short time scale investigated here. Whereas our work stays in the DD condition e.g. of Carr-Purcell (CP) *n* = 2, the authors' choice of $\tau_2$ values in Fig. 3c acts as a filter to probe the two dephasing contributions arising either from nitroxide or solvent nuclei. Though this short time window is less relevant for DEER application work, understanding of this type of decoherence contributions are of fundamental interest. The authors should mention this contribution to also provide a more coherent discussion, as in the context of OX063 and Gd(III) data, dephasing mechanisms arising from the paramagnetic species itself are discussed (see next point).

Thank you for pointing this out. Our first-principle simulations indeed do not fully reproduce the cusp-like feature of the curves in Fig.3c for tau1 = 1 μs, indicating that there is a small contribution from another dephasing mechanism. However, this has no effect on the findings of our work, which show that the optimal tau1 depends on the choice of tau2.

- **Line 212-214:** "This indicates that NSD, induced by the trityl OX063 protons themselves is still a contributing mechanism …". I note that our own work also identifies the OX063 protons as a source for NSD and we compare its decay under DD to the partially deuterated trityl radical OX071, demonstrating the DD is more efficient for short interpulse delays (DD-watergly2021: Section 3.3.3 in the main text, and Fig. Sa20 b/d in the SI part A) compared to partial trityl deuteration. Hence, we have experimental evidence for what is a speculation here.

We are glad that JS21 includes evidence for this. We now mention this and refer to JS21, see line 220. If you wish, you can also refer to our results in your paper when published.

- **Line 218-219:** "In protonated solvent, the shape of the 2D decay is similar to the ones observed for 3-maleimido-proxyl and trityl OX063". I note that the echo maximum in Fig. 7a does not follow the $\tau_1 = \tau_2$ line even at short delay times, whereas it does follow this line at early times for nitroxide/trityl in Fig. 3a/6a. Therefore, the behavior should not be called "similar". It likely originates from a ZFS-driven dephasing contribution (see next point).

We will modify the sentence as follows "In protonated solvent (Fig. 7a), the shape of the 2D decay is generally similar to the ones observed for 3-maleimido-proxyl and trityl OXO63, except for short $\tau_1$ and $\tau_2$, where the slice-wise echo maxima detuned from the CPMG condition (red lines)."

- **Line 220-222:** "…, indicating that a second dephasing mechanism in addition to NSD is contributing, such as the transient zero field splitting mechanism. (Raitsimring et al., 2014)"

  **Line 223-225:** "in deuterated solvents, … nuclear spin diffusion plays a role [for Gd(III)] dephasing that is lower than in trityl OX063 but higher than in 3-maleimido-proxyl"

  This discussion is inconsistent and requires further elaboration regarding the transient zero field splitting (tZFS) mechanism/Raitsimring et al., 2014 for the following reasons. First, the cited work introduces the tZFS for $|m_S| > \frac{1}{2}$ transitions. Specifically, the Abstract of the article states "tZFS induced phase relaxation mechanism becomes dominant (or at least significant) when all other well-known phase relaxation mechanisms, such as spectral diffusion caused by nuclear spin diffusion, instantaneous and electron spin spectral diffusion, are significantly suppressed by matrix deuteration and low concentration", and the cited article furthermore argues that the $|m_S| = \frac{1}{2}$ transition behaves analogous to a $S = \frac{1}{2}$ system, meaning that the dephasing at this field position is NSD-driven. Based on this citation alone, it is astonishing that the authors consider the tZFS mechanism in case of the protonated solvent but do not discuss its contribution in the deuterated case. Moreover, as the refocused Hahn echo decay was recorded at the maximum of the EPR spectrum, thus probing the $|m_S| = \frac{1}{2}$ transition, the authors need to comment on this finding as it is in conflict with the statement in Raitsimring et al., 2014. In fact, our own DD study relies on three Gd(III) complexes with varying ZFS to demonstrate that in protonated water-glycerol glass a ZFS-driven dephasing mechanism contributes at the central Gd(III) field position (DD-watergly2021: Fig. 7a, Section 3.4.2). Second, the authors' spin concentration choice of 100 μM is highly likely to lead to ID contribution in deuterated water-glycerol glass. For this reason, the observed decoherence behavior (Line 223-225) cannot be interpreted as deuteron-driven NSD exclusively. On the one hand, because tZFS dominates the dephasing for Gd(III) (as our own DD data proves, DD-watergly2021: Fig. 7c). On the other hand, because at the same spin concentration and pulse excitation bandwidth, ID provides a more significant dephasing pathway for trityl radicals compared to nitroxides. We discuss this aspect in our own work (DD-watergly2021: section 3.3.1), stating that nitroxides are spectrally more diluted. To assess differences in deuteron-driven NSD, the authors would need to choose a lower spin concentration and demonstrate experimentally that ID is negligible.

Thank you for this detailed discussion. Our intent is not to provide an in-depth discussion of all relaxation mechanism in all samples but rather our focus is on (partially) protonated samples and the fact that the optimal setting of tau1 and tau2 deviates from the CP condition if tau2 is not too short.

We clearly show the proton NSD is very active for Gd(III) in protonated solvents. It was not our intent to make specific statements about deuteron NSD, since we don't have it

isolated well experimentally and also because we don't have a first-principles theoretical handle on it.

To clarify this, we will add in line 235 "As evident from Fig. 7c, proton nuclear spin diffusion, arising from protons on the Gd(III) chelate, plays a role in dephasing that is lower than in trityl OXO63 (Fig.6c) but higher than in 3-maleimido-proxyl (Fig. 4c)."

**Comments regarding the data analysis/CCE simulations:**

- **Fig. 3a/b** seem to display a small asymmetry with respect to fixed $\tau_2$, variable $\tau_1$ compared to fixed $\tau_1$, variable $\tau_2$. The same appears in Fig. 6a for OX063. The authors should comment whether this is an artefact arising from the data analysis or reflects a true asymmetry.

  This is an experimental imperfection. Theoretically, in the high-temperature limit (which is applicable here), with ideal pulses (neglecting intra-pulse evolution), and for $T_1 \gg T_M$ (applicable here), one can show that $V(tau1,tau2) = V(tau2,tau1)^*$.

- **Figure 5a/b:** The location of the maxima along $\tau_1/\tau_2$ (red lines) display many irregularities in particular for small interpulse delays. The authors should comment on their origin. Potentially, these stem from $^2H$ ESEEM, if this is the case, the authors should specify in the Methods section how these modulations are treated during the normalization.

  These wiggles are a consequence of noise in the experimental data. For small inter-pulse delays, the echo decay along tau1 for a given tau2 (and vice versa) are relatively flat, so that experimental noise can generate larger apparent scatter of the ridge points.

- **Line 260:** The presented CCE simulations are performed in a water-glycerol mixture, though previous published CCE results were obtained in pure water (Canarie et al., 2020), reasoning that "since MD simulations in pure water are well calibrated, whereas water-glycerol mixtures are significantly less tested against experiment, particularly in the solid phase." How did the authors ensure that the calibration of the water-glycerol glass is appropriate?

  Water and water/glycerol have very similar proton concentrations, so the decays are expected to be similar. We now have simulations that compare water and water/glycerol mixtures that show that the two matrices give very similar results. In the below figure, the water/glycerol simulation is the 3-CCE simulation from Fig.10. The water simulation has the same parameters. There are 494 1-clusters, 4124 2-clusters, and 49178 3-clusters. We will mention this in the text and add it as Fig. S3 to the SI.

[Figure]

- **Line 275:** "Remarkably, [the simulated refocused-echo decay] matches the experimental result both in shape and time scale…"

The reader would benefit from adding the experimental data trace of CP $n$ = 2 in Fig. 9c so that the time scale and shape as well as CCE convergence can be judged more easily (e.g. as done in Canarie et al., 2020). This display is likely to reveal a mismatch for short $\tau_1/\tau_2$ as evident from comparing the normalized slices (red lines) in Fig. 3a-b with the ones in Fig. 9 a-b. I also expect this from my own experimental results (DD-watergly2021: SI, Fig. Sa12 CP $n$ = 2 for protonated nitroxide in protonated water-glycerol at 40 K – according to Fig. Sa5 comparable to decay behavior at 20-30 K used in the article under discussion). This contribution originates from the methyl protons of the nitroxide.

The figure below shows the suggested plot (color: experiment; dashed black: first-principle simulation). It shows the first-principle simulations are overall in remarkable agreement with experiment. It also illustrates the mentioned deviation of the theory from experiment at short tau2. We will mention this in the text and add this Figure to the SI as S4.

[Figure]

- **Line 281-282:** "Including two-nucleus clusters in the simulation yields an echo decay that has the correct shape and an almost correct time scale. Adding three-nucleus clusters improves the time scale slightly…" Considering Fig. 9c 2-CCE is ~ 1 μs off from

3-CCE decayed at 4 μs. This deviation is relatively large and convergence appears to be reached for 3-CCE so that the authors should reconsider their somewhat misleading wording here.

We will reword this from "almost correct time scale" to "time scale of the correct order of magnitude."

**Comment regarding the sample choice:**

The article presents the refocused echo decay as a function of $\tau_1$ and $\tau_2$ for a nitroxide, a trityl radical and GdCl3 in protonated and deuterated water-glycerol glass as specified in the Introduction (Line 78-79). The authors should justify the additional sample choice/discussion of the Gd-C2-labeled MdfA protein solubilized in detergent (DDM) micelle without providing the full data with $\tau_1$ and $\tau_2$ variation. First, because this sample varies many experimental parameters at once, namely

- Gd-C2-complex instead of Gd(III) ion, altering the ZFS
- additional HF field arising from the protein's protons (which appears to be the variable of interest and thus should be the only varied parameter)
- micelle environment instead of aqueous water-glycerol mixture
- two labeling sites which may be exposed to different local environments

Second, compared to frozen water-glycerol solvents the micelle environment is known to accelerate the electron spin dephasing strongly (*e.g.* see Dastvan et al., 2010). For both reasons, it is not clear to me how the reader benefits from this somewhat unconnected "application example".

The point of this application example is to show that there are cases where even when the solvent is fully deuterated, there are decoherence contributions from the remaining proton bath, and one can choose a long tau1 for collecting the data. So to our opinion, this is a practically relevant example that complements the more fundamental exploration of the other samples. See also our response to a similar comment from Reviewer 3.

For a stronger discussion, the authors should consider to compare a single-labeled water-soluble protein with the chosen spin label in the same solvent environment e.g. at best in a deuterated water-glycerol mixture to be sensitive to the protein's protons. Our own DD study in water-glycerol took exactly this approach for Gd-DOTA-M (DD-watergly2021: Fig. 7c-d and section 3.4.2), demonstrating the decoupling effect for proton-driven NSD arising from the protein's backbone.

While we think this is an interesting and worthwhile comparison, it is not within the scope of this manuscript. We will cite JS21 in this regard.

**General comments:**

- Line 250: "range of 3-4 µs" (blue trace)? This should refer to the yellow and purple trace.

  Fixed.

- Due to the eight-step phase cycle your experiments do not correspond to a CPMG but instead to a CP sequence.

  We will change all CMPG occurrences to CP in our manuscript when referring to our measurements.

**Comments regarding citations:**

- **Line 45/Eq 1:** Zecevic et al., 1998 uses the stretched exponential model, but the cited equation cannot be found in this work.

  The equation is a combination of Zecevic 1998 (which omits T1, V0, and lambda) and Jeschke/Polyhach 2007 (which omit the stretched exponential and lambda and V0), and our addition of V0 and lambda, which are generally known prefactors.

  We will add Jeschke/Polyhach to reference the T1 factor.

- **Line 50:** In Jeschke and Polyhach, 2007 the approximation reads $\tau = \tau_2$ if $\tau_2 \gg \tau_1$ and in this limit the Hahn decay approximates the refocused echo decay well (as visible in Fig. 3c, Fig. 4c, Fig. 6c). If the statement in Line 72-75 "In the context of DEER, it is usually assumed that the refocused echo decays monotonically as a function of the overall pulse sequence length $2(\tau_1 + \tau_2)$, similar to the two-pulse echo." refers to the above approximation, it should be rephrased.

  See corresponding comment to reviewer 3.

- **Line 132-133:** Technically, Harbridge et al., 2003 determined the CPMG time constant, which corresponds to the decay of the n refocused echoes between n refocusing pulses. The work under discussion observes the decay of the refocused Hahn echo, more closely related to our dynamical decoupling (DD) study in OTP (Soetbeer et al., 2018) as well as our recent DD study in water-glycerol glass (DD-watergly2021). Both works systematically address the effect of DD for nuclear spin diffusion (NSD) for organic radicals (and gadolinium complexes) "dilute frozen solutions at cryogenic temperature" (Line 131-132) for both protonated and deuterated matrices, the authors should cite.

We now added Soetbeer, 2018 as well as JS21.

- **Line 177-179:** "It is apparent that NSD is suppressed here and DD is ineffective. The decay is dominated by other dephasing mechanisms such as instantaneous diffusion (ID) …" We demonstrated this effect for 20 compared to 100 µM protonated nitroxide in deuterated OTP (Soetbeer et al., 2018, Fig. 8d-f). The latter matches the used concentration choice in the present work, so that a citation would be appropriate.

  The effect of concentration on the instantaneous diffusion rate has been known for a long time. For completeness, we will add the mentioned reference.

**References**

DD-watergly2021        https://doi.org/10.26434/chemrxiv.13678447.v1

Canarie et al., 2020        https://doi.org/10.1021/acs.jpclett.0c00768

Soetbeer et al., 2018        https://doi.org/10.1039/C7CP07074H

Raitsimring et al., 2014 https://doi.org/10.1016/j.jmr.2014.09.012

Dastvan et al., 2010        https://doi.org/10.1021/jp1060039

---

## Author Comment (AC2)

**Reviewer 1**

Thank you for your positive evaluation and constructive comments. Below we copied our evaluation in black and we present our response is in red.

The manuscript "The decay of the refocused Hahn echo in DEER experiments" is a significant advance toward understanding nuclear spin diffusion and its role in limiting most types of pulse EPR spectroscopic experiments. Nuclear spin diffusion and other mechanisms contributing to spectral diffusion and decay of signals limit sensitivity and the length of time for which a signal can be measured. Efforts to model it have been made since the 1960's, but required oversimplification of the model to such an extent that in many cases echo decay would be impossible or results were qualitative.

However, this paper uses computational power and modeling techniques that are now available to treat the spin system and spin-spin interactions without oversimplification and to construct realistic molecular models of the distribution of nuclei in the sample. The result is an impressive quantitative agreement with experimental measurements in three different systems relevant to many DEER experiments. This provides some insights and guidance on how to optimize samples and measurements. However the model applied here also has some relevance to other pulse EPR measurements such as: ESEEM, ENDOR (both Mims and Davies), and HYSCORE, to name a few. This paper has relevance and impact for other forms of pulse EPR.

The experimental part of the paper and the choice of samples are a good compromise between freedom from other sources of echo decay and relevance to typical DEER measurements. So results at the longest times and for the highest deuteration may be limited by appearance of instantaneous diffusion, local modes, molecular motion, and methyl group rotation. But within those boundaries, the calculations and experiments seem in good agreement.

Measurements were also made of a Gd-labelled protein. There are many grounds for criticizing the use of this particular sample. It certainly cannot be used to validate the modeling and calculations. However, it provides an important indication that the results, that are validated in better defined model systems, do have relevance to 'real' samples.

We agree that the main purpose of the protein sample is to show that the results are relevant for biological samples.

Although it is not really mentioned in the paper, one of the important aspects of the experimental measurements is that they are made at W-band. This almost completely suppresses any ESEEM from protons and deuterons both because of its tiny amplitude at high magnetic field and because of the difficulty in exciting it with microwave pulses broader than the nuclear Zeeman period. Labs operating at lower microwave frequencies will be affected by ESEEM but the computations as described here also

would include ESEEM. The point is that ESEEM becomes relevant at lower frequencies and may modify the results obtained here for W-band, but that point lies beyond the scope of this paper in establishing the modeling and calculations.

We agree with the reviewer that ESEEM is more significant at lower microwave frequencies. In our previous work (Canarie et al., J.Phys.Chem.Lett., 2019), we have already investigated the field dependence of the echo decay and have shown that the nuclear-spin-bath-driven decoherence is field independent, whereas the ESEEM modulation depth is field-dependent. The CCE simulations in the following figure illustrate this by way of a comparison between Q-band and W-band. The initial parts of the decay differ due to different ESEEM modulation depths, but the tails as well as the overall time scale of the decay remain unaffected. This will be added to the manuscript with a reference to the figure which will be as supplementary information as S2.

[Figure]

However, the paper does not disclose some very important and relevant experimental details needed for readers to evaluate the experimental results. What are the approximate pulse widths and turning angles of the microwave pulses in the measurements?

Both the pulse widths and the turning angles are given in Table 1.

Does the strength of the perpendicular part of the microwave magnetic field vary across or along the samples?

Rabi nutation curves show several oscillation periods, from which we conclude that the $B_1$ inhomogeneity is small. We did not evaluate this quantitatively, as there is also an effect of the resonance offset. Also, in our simulations the observed decay time scale is independent of the pulse flip angles.

Were any checks made for instantaneous diffusion at the longest times?

The instantaneous-diffusion decay constant is 80 µs at 100 µM bulk concentration and a 25% flip probability. Its effect on the echo intensity for the time window where the echo is significant in our samples (5 – 10 µs) is therefore minor.

What was measured--peak point of echo, integral of echo, window between half height points of echo,...?

The echo was integrated over its full width at half maximum, as noted at the end of the "Spectroscopic measurements" paragraph in the Methods section.

Although it is possible to find many things that could have been added to this paper, they do not seem to reach the importance of two major results: 1) a framework for quantitatively modeling the effect of nuclear spin diffusion on pulse EPR measurements; 2) confirming the importance of pairs and triples of nuclei in nuclear spin diffusion-driven electron spin echo decay.

We agree with this assessment. These are the two major points in this work.

I did find a couple of typos that need correcting: line 279 - "couplings IS neglected"; and line 351 - "socalled".

Thank you – fixed.

The chapter by Ian Brown should be supplemented by the chapter (W. B. Mims, in Electron Paramagnetic Resonance, ed. S. Geschwind, Plenum, New York, 1972, pp. 263-352.) and by the book on spin echoes by Salikhov, Semenov and Tsvetkov (or perhaps the chapter by Salikhov and Tsvetkov in Kevan and Schwartz, I think it covers nuclear spin diffusion).

Thank you for pointing them out, these are indeed important to refer to in the context of nuclear spin diffusion. We will add these references.

---

## Author Comment (AC4)

**Reviewer 2**

Thank you for your positive evaluation and constructive comments. Below we copied our evaluation in black and we present our response is in red.

The authors describe in this manuscript the optimization of pulse settings (the original Hahn echo time) of a 4-pulse DEER experiment. Experiments are shown for nitroxide, trityl and Gd paramagnetic centers at W-band frequencies for deuterated and protonated water solvent. They demonstrate that the time interval t1 optimally is optimally set as t2 for short times of t2 and somewhat shorter for long t2 values. These experimental findings could be quantitatively reproduced by computer simulation of the interaction between the radical with 2-5 nuclear spins - similar to an earlier publications where such calculations have been demonstrated for experiments at more common Q- and X-band frequencies. Overall this work demonstrates very impressively that quantitative simulation of the experimental data can be achieved with coupled spin clusters (of at least 3).

I have some questions and remarks which might be helpful also for other readers in the final version of the paper:

1) The data are all recorded at W-band. This has of course the advantage that the classical ESEEM effects are reduced. On the other side the authors describe the decoherence of the EPR signal as a nuclear-spin-cluster electron spin decoherence arising from the interference of nuclear coherences arising from the different hyperfine coupling of two nuclear spins. This is an interesting model but would probably also imply a field dependence. It would be interesting to give a statement in this direction.

In our earlier work (Canarie et al., J.Phys.Chem.Lett., 2020), we have shown that the effect is field independent (no change between X and Q band). Also, theoretical considerations of a simple system of one electron spin and two spin-1/2 nuclei (see Lenz et al, ChemComm, 2017) shows that the effect is independent of the field and depends only on the ratio of the nucleus–nucleus coupling to the difference in the two hyperfine couplings.

Below are simulations for W-band versus Q-band. They show that the only difference is in the ESEEM modulations, and that the nuclear-spin-cluster driven dephasing is field independent:

[Figure]

We will add a statement that refers to the field independence of the decoherence effect and add this figure as S2 to the SI.

Also, the term SzAzxIx is omitted in this calculations (different from the earlier publication for data at Q-band and X-band). It would be good to also explain this in more detail.

The SzIx is in fact included in the numerical simulations in Fig. 9 that reproduce the experimental data, and also in the simulations in Fig.10, with the single exception of the top left panel (the <V2> LCE term).

We will clarify this better in the Methods section.

The experiments are performed at the maximum of the EPR spectra. Tm for nitroxides at high field is known to be orientation dependent. Again the authors should comment on this aspect.

We carried out similar measurements at a different field position ($g_{zz}$) and did not see any differences. We will mention this on p. 7 and add a Figure in the SI. On line 168 we will add "We also checked this behavior by setting the magnetic field to the region of $g_{zz}$ and saw the same behavior (see Fig. S1)."

2) The authors demonstrate that for samples with 25% 50% 75% and 100% protonation this dephasing is efficient but claim that this is not the case for fully deuterated samples. In the cited work by Soetbeer et al. it is shown that dynamic decoupling is also effective for 100% deuterated samples. It would be nice to discuss this point more carefully.

The relevant comparison is with H-NO in $D_2O$/glycerol. We did not find DD data on this sample in the ChemRxiv preprint by Soetbeer et al, 2021. There are data on the D-mNOPEG, where the spin labeled is attached to a polymer with a significant number of protons. Similarly, in Soetbeer et al 2018, they looked at nitroxides attached to a molecule with many protons. These will contribute to dephasing which can be refocused by DD. We discuss this in lines 180-184.

Also in the above mentioned work Tm was analysed by two different components with stretched exponential with a 50/50 ratio. It would again be important to discuss this differences to the treatment here which is relying only on one mechanism.

Although in this work we focused on the decay of the refocused echo as a function of the time intervals $\tau_1$ and $\tau_2$, it is interesting to compare our Hahn echo decay shapes with those reported earlier by Jeschke and co-workers, where the data were analysed in terms of a sum of two stretched exponential, one with a fast decay and another with a slow decay. We do not observe the fast decay, which was particularly prominent in the protonated and deuterated nitroxides in $D_2O$/glycerol-$d_8$ at low temperatures (10-50 K). Similarly, this was not observed in our earlier report on trityl, nitroxide and Gd(III) spin labels, free and attached to a protein. The difference could be due to the field (Q-band vs W-band) or the different type of nitroxide used.

We will add this to the discussion (line 365). Although we do not see in the echo decays the fast relaxation, the small cusp observed for short tau2 may be an indication for a small contribution of another mechanism, but we did not investigate further because it did not affect the position of the maximum.

3) The red maxima shown in the 2D datasets should also be presented in 4 6b and 7b for consistency.

These mentioned figures refer to deuterated solvents. In these, the red maxima would be coinciding with the coordinate axis, since the effect observed in protonated solvents is absent. Therefore, we have omitted these lines.

---

## Author Comment (AC5)

**Reviewer 3**

Thank you for your positive evaluation and constructive comments. Below we copied our evaluation in black and we present our response is in red.

This is a very interesting study by the Goldfarb and Stoll labs demonstrating that the common assumption of short tau1 values leading to larger signals in DEER might not always be met. Tau2 will determine the distance range that can be retrieved from the DEER data and tau1 is commonly chosen short to minimize time for echo dephasing. The authors very clearly demonstrate that extending tau1 for a given tau2 can lead to increased sensitivity. This appears to be most relevant for samples with limited possibilities for deuterium exchange. Nevertheless, this is an important finding to report especially as optimizing tau1 for a given tau2 will likely be a very quick experiment in contrast to DEER averaging times that will often average for many hours if sensitivity is limiting. The authors further make an excellent effort to rationalize their findings in in terms of numeric simulations and conceptualization.

From the practitioner's point of view this has sparked a number of questions that might be worth commenting on in the final version of the manuscript. I am aware that some of the simulations or experiments that would be required to exhaust these questions will be beyond the scope of this work but I believe at least commenting on them will be of interest to the reader.

All experiments and simulations are performed at W-band. Considering that most reported DEER experiments have been measured at X- and Q-band how do these effects translate at lower fields. I suppose the non-zero transition amplitudes of the formally forbidden transitions will increase while the nuclear Larmor frequency will decrease. Is the overall effect field-independent? This should be straightforward to simulate. The title of the manuscript suggests a general treatment.

In our earlier work (Canarie et al, J.Phys.Chem.Lett., 2020), we have shown experimentally and by numerical simulations that the effect is field independent (no change between X and Q band). Also, theoretical considerations of a simple system of one electron spin and two spin-1/2 nuclei (see Lenz et al, ChemComm, 2017) shows that the effect is independent of the field and depends only on the ratio of the nucleus-nucleus coupling to the difference in the two hyperfine couplings.

Below are simulations for W-band versus Q-band. They show that the only difference is in the ESEEM modulations, and that the nuclear-spin-cluster driven dephasing is field independent:

[Figure]

Would softer pulses be expected to lead to decreased dephasing. This has been shown in the context of instantaneous diffusion (Jeschke and Polyhach, 2007) but in terms of forbidden transitions this might be relevant here as well.

Instantaneous diffusion is indeed an additional dephasing mechanism that needs to be considered. We checked for instantaneous diffusion by measuring the Hahn echo decay with softer pulses but we found the decay curves to be identical. This will be mentioned in p. 7 line 170. We will add: "We also checked that the contributions of instantaneous diffusion under these conditions was negligible by comparing the Hahn echo decay obtained with different pulse lengths (see Fig. S1c)".

When deuterating the solution of 3-maleimido-proxyl the data are interpreted as nuclear spin diffusion being suppressed and dipolar decoupling becoming ineffective as other dephasing mechanisms become dominating. Has this been explored using lower concentration or softer pulses? At sufficiently low concentration would dynamic decoupling become effective again in deuterated samples. Could deuterium nuclei be simulated using the same approach but potentially fewer nuclei?

We did not explore the relaxation mechanisms for the deuterated samples. The challenge is that the currently established CCE theory breaks down for a bath of spin-1 nuclei such as deuterium, because the CCE series expansion does not converge at low orders. Theoretical work is under way to work around this impasse, but hasn't been successful yet. In other words, accurately simulating a spin system with one electron spin and a few hundred deuterons is still out of reach.

Different scenarios of residual proton content will likely lead to different outcomes. 25% of protons already have a significant effect but there are no experimental points up to full deuteration. Is it feasible to thin out the protons in the simulation until the dephasing effect will vanish when proton clusters with sizeable nuclear couplings become improbable.

We agree that this would indeed be of theoretical interest. As mentioned in the previous point, however, accurate theoretical simulations for situations of very low proton concentration are currently not possible, since then the deuterium bath starts co-determining the decoherence time scale, and the CCE expansion for a spin-1 bath has

convergence problems. We hope to perform these simulations, and the associated experiments, once we have overcome the theoretical hurdles.

For non-homogeneous distributions of protons that will be most relevant practically (El Mkami et al., 2014) it will be very interesting to see the influence of the proximity of protons. The full effect was recovered with protons in 1.2 nm. This suggests the dephasing of a spin label well solvated in deuterated solution away from the protein will be substantially slower than when buried in the fold of a protonated protein or membrane. Will the simulation approach be applicable to inhomogeneous distributions of protons?

Yes, the simulation approach is fully general and applies to any spatial distribution of protons. There is a very large spread of $T_M$ for nitroxides label on different sites of the same protein. However, accurate comparison of simulation and experiment is more challenging for proteins due to the large structural modeling uncertainty of the solvated protein structure, in particular the side chain conformation of the spin label. In addition, another important relaxation mechanism in proteins is mediated by tunneling in protein methyl groups. This is currently not quantitatively understood, preventing a successful comparison of experiment and simulation for protein samples.

The MdfA double mutant V44C/V307C doubly labelled with Gd-C2 is measured in detergent micelles. Without further knowledge of structure and labelling positions the effect of non-exchangeable protons is hard to predict. An earlier report by Dastvan et al. (https://doi.org/10.1021/jp1060039) suggests the increased proton density in lipids in comparison to aqueous solution leads to increased dephasing. This might also be relevant for detergent. In this light, this might not be the most relevant protein system to demonstrate these results from homogeneous solutions of free spin labels.

Our intent with the MdfA example was precisely to show this. There are cases where even when the solvent is fully deuterated, there are protons that drive decoherence. In such situations it is advantageous to choose a long tau1 for collecting the data, as shown in Fig. 8. Therefore, in our opinion, this is a real practical and relevant example, more so than a soluble protein with surface-exposed spin labels in fully deuterated buffer. To clarify this better, we will reword the corresponding section (see line 256) and also cite the paper by Dastvan et al.

Without having done the simulations, is my extrapolation that a larger number of proton clusters and larger couplings between protons expected for media with increased proton density will lead to faster dephasing consistent with the findings here?

This is indeed correct. The dephasing rate i1/$T_M$ is indeed roughly proportional to bulk proton concentration. This was already found experimentally by Zecevic et al in 1998, by canvassing a range of solvents with different bulk proton densities. For low proton concentrations, the geometric details of the proton clusters will likely have a strong

impact on $T_M$. However, from our numerical simulations we have not been able to identify simple intuitive rules.

Further points

The introduction of the 3 and 4 pulse DEER sequences seems to suggest they were initially reported in 1984 and 2000, respectively. I suggest changing the wording or adding the original references.

Thank you for noting this. We will update the reference for 3-pulse DEER to Milov 1981, and the reference for 4-pulse DEER to Martin 1998.

(Jeschke and Polyhach, 2007) set the S/N ~ exp(-2tmax/T2) and this still holds in the approximation that even with an optimized tau1 the refocused echo will decay exponentially with tau2.

We referred to eq. 3.16 and 3.17, which are more general. Yes, with a fixed tau1 the echo will decay with tau2, but not necessarily exponentially, but with a stretched exponent.

The discussion of dephasing by electron-electron dipolar interaction is confusing. An increased concentration will lead to larger signal and faster dephasing. As shown in (Jeschke and Polyhach, 2007) there will be an optimal concentration depending on the required trace length. If dilution lead to longer averaging times dilution it was overdone.

We fully agree. We should have made this clearer in our discussion. To address this, we will add: "However, this concentration reduction leads to a loss in absolute signal intensity and may significantly prolong the experiment run time and therefore there is an optimal concentration for best SNR."

In line 74 it is said short tau1 values minimize phase relaxation" but considering instantaneous diffusion I suggest "minimize dephasing".

We will adjust this.